# CCL22-Polarized TAMs to M2a Macrophages in Cervical Cancer In Vitro Model

**DOI:** 10.3390/cells11132027

**Published:** 2022-06-25

**Authors:** Qun Wang, Kritika Sudan, Elisa Schmoeckel, Bernd Peter Kost, Christina Kuhn, Aurelia Vattai, Theresa Vilsmaier, Sven Mahner, Udo Jeschke, Helene Hildegard Heidegger

**Affiliations:** 1Department of Obstetrics and Gynecology, University Hospital, LMU Munich, 80377 Munich, Germany; wqyxdz888@163.com (Q.W.); bernd.kost@med.uni-muenchen.de (B.P.K.); christina.kuhn@uk-augsburg.de (C.K.); aurelia.vattai@med.uni-muenchen.de (A.V.); theresa.vilsmaier@med.uni-muenchen.de (T.V.); sven.mahner@med.uni-muenchen.de (S.M.); helene.heidegger@med.uni-muenchen.de (H.H.H.); 2Department of Cardiology, University Hospital Munich, LMU Munich, 81377 Munich, Germany; kritika.sudan@iomx.com; 3Department of Pathology, LMU Munich, 80377 Munich, Germany; elisa.schmoeckel@web.de; 4Department of Obstetrics and Gynecology, University Hospital Augsburg, 86156 Augsburg, Germany

**Keywords:** cervical cancer, TAMs, CCL22, M2a macrophages

## Abstract

Macrophages are dynamic cells susceptible to the local microenvironment which includes tumor-associated macrophages (TAMs) in cancers. TAMs are a collection of heterogeneous macrophages, including M1 and M2 subtypes, shaped by various activation modes and labeled with various markers in different tumors. CCL22+-infiltrating cells are thought to be significantly associated with the prognosis of cervical cancer patients. Moreover, CCL22 is an established marker of M2a macrophages. Although the phenotypic identification of M1 and M2 macrophages is well established in mice and human macrophages cultured in a medium with fetal calf serum (FCS), fewer studies have focused on M2 subtypes. In addition, the question of whether CCL22 affects polarization of M2a macrophages remains unanswered. This study constructed a co-culture system to shape TAMs in vitro. We found that CCL22 was mainly secreted by TAMs but not cervical cancer cell lines. Human peripheral blood monocytes were differentiated into uncommitted macrophages (M0) and then polarized to M1, M2a, M2b, and M2c macrophages using LPS plus IFNr, IL-4, LPS plus IL1β, and IL-10, respectively. Using flowcytometry, we found CD80++ was the marker of M1 and M2b, CD206++ was the marker of M2a, and CD163++ was the marker of M2c, compared with M0 macrophages. By regulating CCL22, we found that the mean fluorescence intensity (MFI) of CD206 in TAMs was significantly affected compared to the control group. Therefore, CCL22 could polarize TAMs of cervical cancer toward M2a macrophages. In conclusion, our study revealed that CCL22 could be a therapeutic target for cervical cancer, which might be because of its role in regulating macrophage polarization.

## 1. Introduction

Cervical cancer is the fourth leading cause of cancer death in females globally and the third most common cancer [1,2]. Squamous cell carcinoma and adenocarcinoma occupy the most histological subtypes of cervical cancer, accounting for 70% and 25%, respectively [3]. Although recent improvements in therapeutic options, including chemoradiotherapy and surgery, have increased the survival rate of patients with primary cervical cancer, the 5-year survival rate of patients with local or distant metastasis is reduced to 50% [4,5]. At present, immunotherapy is a breakthrough cervical cancer treatment. In particular, the monoclonal antibody pembrolizumab—KEYTRUDA^®^—combined with chemotherapy, with or without bevacizumab (KEYTRUDA regimen), has been developed into a drug that the EMA approved for therapy in patients with persistent, recurrent or metastatic cervical cancer who are positive for PD-L1. Listeria monocytogenes inactivated vaccine vector (ADXS11-001) can be genetically engineered to generate the fusion protein of HPV-16 E7 to stimulate the immune response to E7. However, the overall remission rate is low [6,7]. Therefore, efforts to further identify new immune checkpoints is crucial.

Macrophages are roughly classified into two types: M1 (classically activated) and M2 (alternatively activated) [8]. M1 macrophages are stimulated by Th1 factors and reduce tumor growth. Conversely, M2 macrophages are stimulated by Th2 factors and promote tumor growth [9]. Recently, three different subtypes of M2 macrophages (M2a, M2b, and M2c) were described. Although these M2 subsets share some markers and immunosuppressive functions, different subsets are induced by different mechanisms and have diverse physiological functions [10,11]. M2a macrophages are polarized by IL-4 and act in tissue remodeling and reducing inflammation [12,13]. M2b macrophages are stimulated by an Fc-γ receptor plus LPS or IL-1β and act in an antigen presentation that causes Th2 cell differentiation that is detrimental to cancer immunotherapy [14,15]. However, some studies suggest that Th2 cells could work well in eliminating cancer as soon as they are transferred [16,17]. M2c is stimulated by anti-inflammatory stimuli, such as IL-10 or TGF, and is associated with immunoregulation, matrix deposition and tissue remodeling [18]. Studies have revealed that M2 subtypes play a role in tumors. M2a synergistically promote migratory and invasive breast cancer cell responses [19]. M2b could promote hepatoma carcinoma progression [20]. Pellino-1 inhibits tumor growth in vivo by inhibiting IL-10-induced M2c macrophage polarization [21]. TAMs, including M1 and M2 macrophages, are shaped by various activation modes and labeled with various markers in different tumors [22]. TAMs mediate immune tolerance to enhance the effects of some anti-tumor drugs. For example, the clearance of TAMs in hepatoma could enhance the effect of inhibiting angiogenesis and the metastasis of Sorafenib [23]. However, some studies reveal that the Fc γ receptor of TAMs participates in antibody-dependent cellular cytotoxicity. Therefore, the clearance of TAMs will block the response of effector T cells, resulting in decreasing pharmaceutical effects [24]. Regulating tumor-promoting TAMs to anti-tumor types is an essential focus of study.

As a kind of secreted protein, the C-C motif chemokine ligand 22 gene (CCL22) participates in chemotactic activity for various immunocytes, including monocytes and chronically activated T lymphocytes [25,26,27,28]. CCL22 has been proved to act in tumor promotion. For example, CCL22 could attract regulatory T cells (T-reg) into tumor mass and decrease cell immunity in ovarian cancer [29]. High amounts of CCL22 expressed in M2 macrophages enables colorectal cancer to be resistant to chemotherapy [30]. Moreover, the mRNA of CCL22 is expressed higher in cervical cancer tissue than that which is present in normal cervical tissue [31]. In addition, we have demonstrated that CCL22 is associated with cervical cancer prognosis [32]. However, the mechanism of CCL22 to promote cervical cancer progression remains unclear. In this study, we assessed the effect of CCL22 on the expression of M2a marker CD206 by flowcytometry and revealed that CCL22 could be a therapeutic target for cervical cancer, which might be due to its role in regulating macrophage polarization.

## 2. Materials and Methods

### 2.1. Cell Culture

Buffy coats from 10 healthy blood donors were obtained from the Bavarian Red Cross blood service (Munich, Germany), and peripheral blood mononuclear cells were isolated by density gradient centrifugation using Ficoll (Cytiva, Uppsala, Sweden). Monocytes were purified using the method of adhesion to the bottom of cell culture plates (Greiner bio-one, Frickenhausen, Germany). For macrophage differentiation, 2.5 × 10^6^/well monocytes were seeded in six-well plates cultured in RPMI-1640 plus GlutaMAX (Gibco, Germany) supplemented with 10% heat-inactivated human sera AB (PAN Biotech, Aidenbach, Germany), 1% penicillin–streptomycin (Avantor, Griesheim, Germany) for three days to acquire M0 macrophages. M0 were polarized into M1 using E.coli LPS (20 ng/mL) (Sigma, St. Louis, MO, USA) plus rh IFN-r (20 ng/mL) (Peprotech, Cranbury, NJ, USA), into M2a using rhIL-4 alone (20 ng/mL) (peprotech), into M2b using LPS (1 ug/mL) plus IL1β (10 ng/m) (Peprotech), and into M2c using IL-10 (20 ng/mL) (RD system) for 48 h. Cells were harvested from the culture flasks by trypsin treatment. Suspensions of 2.3 × 105/well cells of Siha, and HeLa cell lines were seeded in the inserts of six-well plates for 48 h in the medium of RPMI1640, 10% heat-inactivated human sera, and 1% penicillin–streptomycin. To acquire TAMs, we co-cultured the cervical cancer cells with M0 macrophages for 48 h. To acquire si-RNA-treated TAMs, we first performed transfection to M0 macrophages, and then co-culture treated M0 macrophages with cervical cancer cells for 48 h.

### 2.2. RT-PCR

RNA was extracted using the RNeasy Kit (QIAGEN, Hilden, Germany), and reverse transcription using a kit of Quantitative PCR was performed using TaqMan primer for CCL22 and β-actin genes in at least duplicate wells (LifeTech, Düsseldorf, Germany) on an ABI prism 7500 (Applied Biosystems, Darmstadt, Germany). After normalizing the data with β-actin, the relative gene expression was calculated by considering M0 as a control [33].

### 2.3. Flowcytometry

Cells were harvested by Accutase (PAN-Biotech, Aidenbach, Germany), washed with PBS, and incubated with Fc blocking solution (10% human sera, PBS). Cells were stained with a panel of surface markers: PE-conjugated anti-human CD45 with clone HI30 (eBioscience, Frankfrt am Main, Germany); PerCP-Cy5.5-conjugated anti-human CD80 with clone 2D10 (eBioscience, Germany); APCcy7-conjugated anti-human CD206 with clone 19.2 (eBioscience, Germany), and APC-conjugated anti-human CD163 with clone GHI/61 (eBioscience, Germany). SYTOX Blue dead cell staining (Invitrogen, Darmstadt, Germany) was performed to gate out dead cells. Data were acquired on a BD Canto II using BD FACS Diva software. Cell populations were identified on CD45-positive cells on FLowjo (Tree Star, Ashland, OR, USA).

### 2.4. RNA Interference and Plasmid Construction

To silence CCL22 expression, M0 macrophages were transfected with siRNA specific for human CCL22 (Qiagen, Germany) or the negative control siRNA (Qiagen, Germany) according to manufacturer’s specifications. After 6 h of incubation, the cells were co-cultured with cervical cancer cell line cells. We borrowed the plasmid expressing CCL22 from the group of Prof. David Anz (University Hospital of Munich, LMU, Munich, Germany) containing a human CCL22 isoform (NM_002990) coding sequence (CDS:CATGGATCGTACAGACTCATCCTGGTGTCCTCGTCCTCCTGCTGTGGCGCTTCACAACTGAGGCAGGCCCCTACGGCGCCAACATGGAAGACAGCGTCTGCTGCCGTGTATCCGTTACCGTCTGCCCCTGCGCGTGGTGAAACACTTCTACTGGACCTCAGACTCCTGCCCGAGGCCTGGCGTGGTGTTGCTAACCTTCAGGTAAGGGATCTGTGCCGATCCCAGAGTGCCCTG) directly cloned into pmx-vector. They confirmed all constructs via sequencing.

### 2.5. Statistics

The data are displayed as the mean ± SD. To compare CCL22 content among multiple test groups, we performed a one-way ANOVA followed by a Newman-Keuls test. We used an unpaired t-test to calculate two-tailed *p*-values to estimate the statistical significance of differences between the two groups. All tests were implemented in SPSS 26 (SPSS). *p*-values less than 0.05 were considered significant.

## 3. Results

### 3.1. CCL22 Expression in TAMs of Cervical Cancer In Vitro

To ascertain CCL22 mRNA level in vitro in cervical cancer, we quantified its expression in TAMs, M1, M2a, M2b, and M2c macrophages. The results showed that the mRNA level of CCL22 in TAMs co-cultured with HeLa or Siha cells was significantly higher than in M1 macrophages. It should be noted that, while there is no significance, the mRNA level of CCL22 showed a trend to be higher in TAMs than in M2b and M2c macrophages, while lower in M2a macrophages. Besides, the mRNA level of CCL22 in cervical cancer cell lines alone or co-cultured with M0 macrophages were extremely low (Figure 1). These results suggested that CCL22 was mainly derived from TAMs but not cervical cancer cells in a cervical cancer microenvironment.

### 3.2. CD206++ and CD163++ Are the Marker of M2a and M2c Macrophages, Respectively

The surface molecules are different in macrophages of various subsets, including CD80, CD86, CD206, LIGHT (TNFSF14), TLR4, and CD163. This restricted panel was selected based on published papers [34,35,36]. The expression of cell surface molecules was tested by flowcytometry. The mean fluorescence intensity (MFI) of molecules of FMO samples and unpolarized macrophages M0 are regarded as the cut-off value of marker+, and marker++, respectively. M1 and M2b macrophages had a significantly higher percentage of CD80++ than other groups (Figure 2a,b). M2a macrophages were found to have a significantly higher percentage of CD206++ than all other subtypes of macrophages (Figure 2c,d). LIGHT showed no significant change (Figure 2e,f). M2c macrophages had significantly higher amounts of CD163++ compared to all other subtypes of macrophages (Figure 2g,h). In summary, CD80++ could not distinguish M1 from M2b macrophages, where the M2a macrophage’ marker is CD206++, and M2c macrophage marker is CD163++ (Table 1).

### 3.3. CCL22 Could Polarize TAMs toward M2a Macrophages in Cervical Cancer via an Autocrine Pathway

As described previously, a higher amount of infiltrating CCL22+ cells predicts a poor prognosis in cervical cancer [32]. The CCL22 mRNA level in a local immune microenvironment of normal cervix tissue was lower than that in cervical cancer tissue [31]. CCL22 is a marker of M2a macrophages [37]. Therefore, it is pertinent to ask whether CCL22 might promote tumor progression via the inducement of M2a macrophage generation. To verify this, we performed RNA interference in M0 cells by transfecting siRNA specific to human CCL22 (referred to as si-CCL22) or a negative control siRNA (referred to as sicon-CCL22). Subsequently, we co-cultured the treated M0 with HeLa and Siha cells, respectively. The efficiency of RNA interference was confirmed by real-time PCR analyses (Figure 3a). The expression of CCL22 mRNA in si-CCL22 TAMs was 17% of sicon-CCL22. Co-cultured sicon-CCL22 TAMs and co-cultured si-CCL22 TAMs were isolated from the co-cultured cells. We next detected the expression of CD206 of siCCL22-TAMs compared to sicon-TAMs by flowcytometry. As displayed in Figure 3b–e and Table 2, knockdown of CCL22 expression greatly decreased the MFI of CD206 of TAMs co-cultured with HeLa and Siha cells, respectively. These data show that knocking down CCL22 prevents the polarization of M2a macrophages. The efficiency of CCL22 vectors was confirmed by real-time PCR analyses (Figure 3f). The expression of CCL22 mRNA in T-CCL22 TAMs were isolated form the co-cultured cells. We next detected the expression of CD206 of T-CCL22-TAMs compared to con-TAMs by flowcytometry. As displayed in Figure 3g–j and Table 3, over-expressed CCL22 greatly elevated the MFI of CD206 of TAMs co-cultured with HeLa and Siha cells, respectively. These data show that over-expressed CCL22 prevents the polarization of M2a macrophages.

## 4. Discussion

Previous human macrophage studies have particularly focused on identifying functional surface markers of M1 and M2 [38,39,40]. However, the conditions used to culture human macrophages have varied in different studies. Meanwhile, macrophages are dynamic cells that are susceptible to the local microenvironment [41,42,43,44,45]. Moreover, only a few studies have been performed on the characterization of M2 macrophage subtypes. Therefore, a clear phenotypic characterization of human M1 and M2 subsets cultured in conditions close to a real human microenvironment is essential for a better understanding of their biological functions and roles in diseases. Here, we present the results of a systematic study analyzing the definitive separation of human macrophages into M1 and M2 subtypes using surface markers. Furthermore, we verify that CCL22 could polarize TAMs of cervical cancer into M2a macrophages.

Among human macrophage studies, the markers are variously caused by different cell types for generating macrophages such as monocytic cell lines or primary cells from blood, different culture conditions such as fetal bovine serum (FBS) or human sera, and different stimuli for polarization. For example, M1 macrophages could be stimulated by both IFNr alone or LPS plus IFNr [46]. However, few studies have been performed in a medium with human sera. Our study uses peripheral blood derived macrophages, 10% human sera in RPMI1640, to imitate a more realistic microenvironment. This method has recently been proved to be a simple and cost-effective method to prepare monocyte derived macrophages with typical morphology representations and applies to in vitro functional studies of macrophages [47]. Macrophage response is highly sensitive to culture conditions. Our study polarizes M1, M2a, M2b, and M2c macrophages with LPS plus IFNr, IL-4, LPS plus IL1β, and IL-10, respectively, which are the most common stimuli to polarize macrophages. Zizzo G et al., using GM-CSF to differentiate human monocyte-derived-macrophages (MDMs) from M0 macrophages, polarized M0 macrophages to M2a and M2c with IL-4 and dexamethasone, respectively. They showed that M2a is CD206+CD163−, while M2c is CD206+CD163+, which is inconsistent with our results [48]. Virginia MS. et al., using RPMI supplemented with octaplas SD, and M-CSF to differentiate MDMs to M0 macrophages, and polarizing M0 macrophages to M2a and M2c with IL-4, IL-10 plus M-CSF, respectively. They showed that CD206 expression is characteristic of M2a, and CD163 expression is characteristic of M2c primed macrophages, which is consistent with our results [49]. Salma I. et al., differentiated human MDMs with DMEM media supplemented with FBS and recombinant M-CSF, and polarized M1 macrophages with IFN-γ, M2a macrophages with IL-4, M2b macrophages with LPS plus IL-1β, and M2c with IL-10. They showed that CD80 is expressed highly in M1 and M2b macrophages, while CD163 is expressed highly in M2c macrophages, consistent with our results [50]. Jelena G et al., differentiating human MDMs with FBS plus M-CSF, and polarizing M2a with IL-4 plus IL-13, showed higher expression of the CD206 marker, which is consistent with our results [51]. Federica R. et al., using RPMI1640 supplemented with heat-inactivated FCS in the presence of M-CSF and polarizing M1 and M2a by LPS and IL-4, respectively, showed that CD80 and CD206 expression in M2a compared to M1 macrophages, both in terms of percentage of positive cells and MFI, were significantly decreased and increased, respectively. This is consistent with our results. Moreover, they found that CD80, which is detectable in M0 macrophages, is associated with high expression of the mannose receptor, CD206, in about 30% of cells [52]. In our study, about 7% of M0 macrophages expressed CD80, and 98% of M0 macrophages expressed CD206, which suggests M0 macrophages are not only differentiated but also polarized to M2-like macrophages in a medium with human sera. However, using our gating method, CD206++ could distinguish M2a from other subtypes of macrophages, including M0 macrophages.

The cellular source of intra-tumoral CCL22 has been a controversial issue. Some studies argue that CCL22 is mainly derived from tumor cells [53]. Other reports show that CCL22 is mainly from tumor-infiltrating cells, such as dendritic cells (DCs) or macrophages [54,55]. Nevertheless, more recent data have provided evidence that in many types of cancers, immune cells are the exclusive producers of CCL22 [55,56]. Previously, by performing a tissue microarray (TMA) using immunohistochemistry (IHC) and double immunofluorescence, we have proved that CCL22 is mainly secreted by infiltrating macrophages and less so by cervical cancer cells in cervical tumor tissue [32]. In this study, by performing RT-PCR, we found that CCL22 was secreted exclusively from TAMs but not cervical cancer cells. The reason for causing the different results might be the use of different experimental samples, cervical cancer cell lines and cervical tumor tissue, and different experimental methods, which have different levels of accuracy.

Many studies have proved that TAMs could promote tumor progression by secreting CCL22. For example, the level of serum macrophage-derived CCL22 is associated with glioma risk and survival period [57]. The ratio of CCL22-positive macrophages to total macrophages was significantly associated with the degree of YK classification in togue squamous cancer [58]. M2 macrophages decreased the inhibitory effect of 5-fluorouracil (5-FU) on colorectal cancer cells migration and invasion by secreting CCL22, and declined the apoptosis induced by 5-FU [30]. Many studies have clarified the mechanism of CCL22 to act in tumors. For example, Curiel and colleagues provide evidence that T regs migrate to and accumulate in the tumor tissue in a CCL22-dependent manner, contributing to an immunosuppressive environment [29]. CCL22 could assist PGE2 in instructing DCs to attract T regs [59]. However, by performing flowcytometry, we are the first to show that knocking down CCL22 significantly decreased the MFI of CD206 in TAMs of cervical cancer compared with the control group, which indicated that CCL22 could polarize TAMs to M2a macrophages. According to published papers, M2a macrophages have a higher angiogenic potential than other subsets [60]. Moreover, the promotion of M2a macrophage attenuates the mutagenic inflammation before tumor initiation [61]. However, regarding our study, further studies need to be performed to show the effects of M2a macrophages on cervical cancer cells after knocking down CCL22.

## 5. Conclusions

Our study has demonstrated that CCL22 was secreted from TAMs but not cervical cancer cell lines. Moreover, in cervical cancer, when macrophages are cultured in a medium with human sera, CD206++ is the marker of M2a macrophages, CD163++ is the marker of M2c macrophages, and CD80++ is the marker of M1 and M2b macrophages. CCL22 could polarize TAMs toward M2a macrophages in cervical cancer. Thus, CCL22 may be a therapeutic target for its treatment.

## Figures and Tables

**Figure 1 cells-11-02027-f001:**
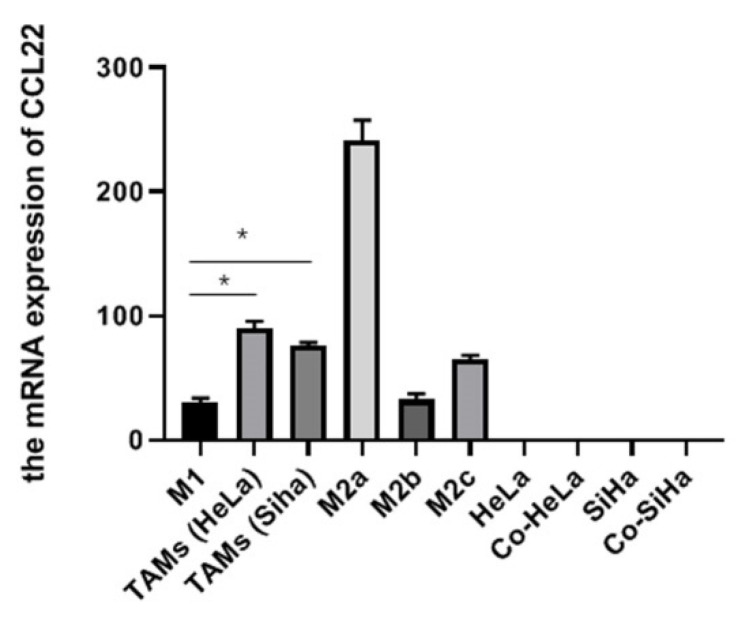
CCL22 expression in TAMs of cervical cancer in vitro. Total RNA was prepared from polarized macrophages (M1, M2a, M2b, M2c), TAMs (tumor-associated macrophages, formed by co-cultured with HeLa or Siha cells), cervical cancer cells HeLa, Siha, co-HeLa, and co-Siha (alone and in co-culture systems). RT-PCR analysis was constructed using CCL22-specific primers. The results are representative of three independent experiments. * Represents *p* < 0.05.

**Figure 2 cells-11-02027-f002:**
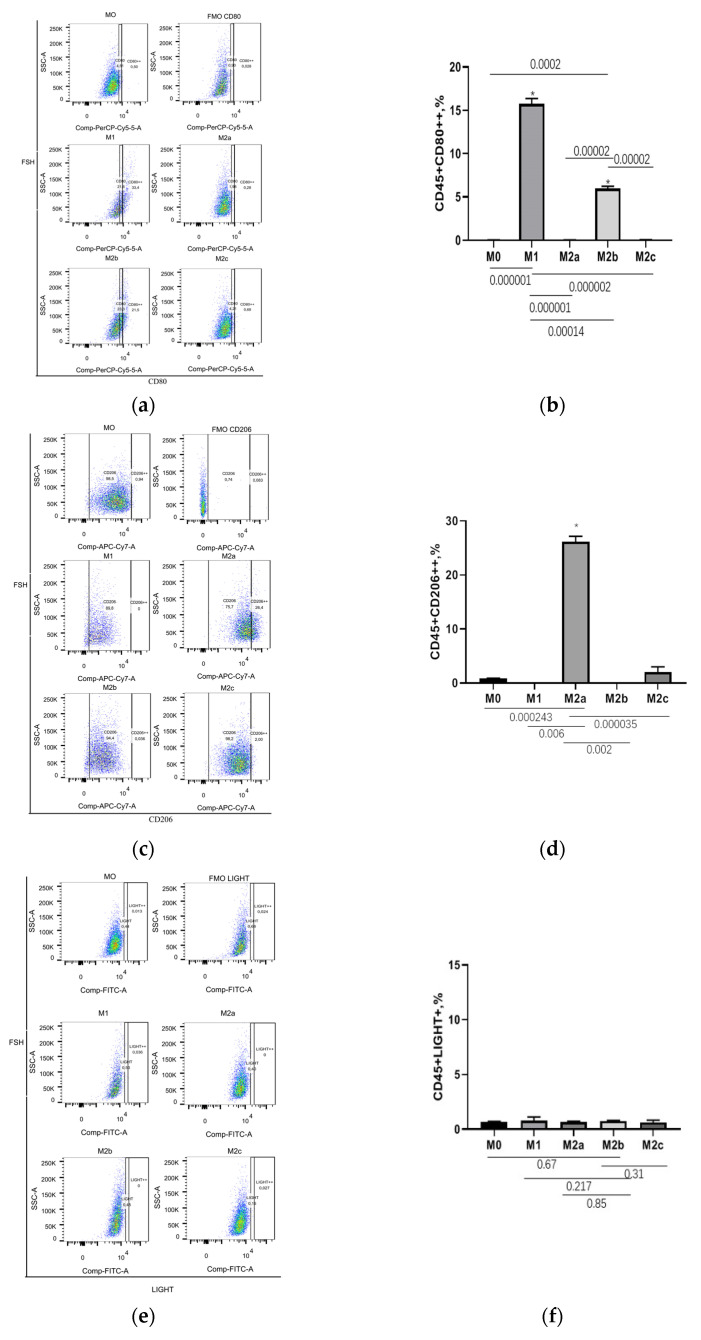
CD206++ and CD163++ are the markers of M2a and M2c macrophages, respectively. Monocytes from healthy donors were cultured in medium with 10% human sera for three days, to differentiate into M0. M1, M2a, M2b, and M2c macrophages were then generated by being stimulated with LPS and IFNr, IL-4, LPS and IL-1b, and IL-10, respectively, for 48 h. Population frequencies of M1 conditioned cells were assessed with two marker sets: CD45 and CD80 (**a**,**b**). Population frequencies of M2a-, M2b- and M2c-conditioned cells were assessed with two marker sets: CD45 and CD206 (**c**,**d**), CD45 and LIGHT (**e**,**f**), CD45 and CD163 (**g**,**h**), respectively. (**a**,**c**,**e**,**g**) are representative flow cytometric plots of the mixture of 2 out of 10 donors. Bar graphs (**b**,**d**,**f**,**h**) represent the mean and SD of population frequencies of CD45+CD80++, CD45+CD206++, CD45+LIGHT+, and CD45+CD163++, respectively, for at least ten different individual subjects. Statistical significance was calculated using nonparametric one-way ANOVA. * Represents *p* < 0.05.

**Figure 3 cells-11-02027-f003:**
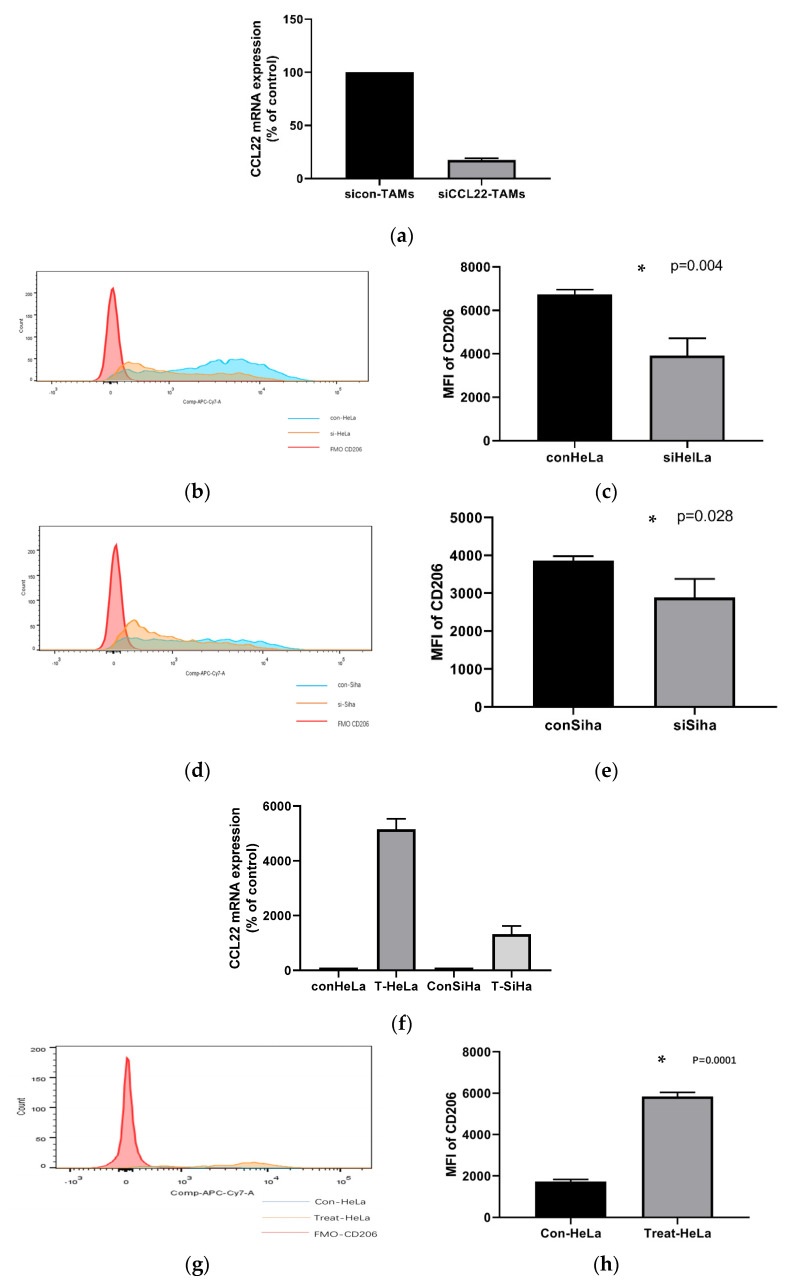
CCL22 could polarize TAMs towards M2a macrophages in cervical cancer via an autocrine pathway. Total RNA from si-CCL22-TAMs and con-TAMs was analyzed by real-time PCR in triplicate for CCL22 mRNA expression (**a**). Flowcytometric analysis of MFI (mean fluorescence intensity) of CD206 on the surface of CCL22 silenced cells (si-CCL22) TAMs and unsilenced controls (con-TAMs) co-cultured with HeLa and Siha cells (**b**,**d**). Total RNA from CCL22-vectors treated TAMs and con-TAMs was analyzed by real-time PCR in triplicate for CCL22 mRNA expression (**f**). Flowcytometric analysis of MFI (mean fluorescence intensity) of CD206 on the surface of CCL22-vectors-treated TAMs and con-TAMs co-cultured with HeLa and Siha cells (**g**,**i**). Red line represents MFI of FMO CD206 group; blue line represents MFI of control CCL22 group; orange line represents si-CCL22 or CCL22-vectors treated group. Bar graphs (**c**,**e**,**h**,**j**) represent the mean and SD of MFI of CD206 in treated macrophages co-cultured with cervical cancer cell line HeLa and Siha, respectively, for at least three independent experiments. * Represents *p* < 0.05.

**Table 1 cells-11-02027-t001:** Population frequencies of human uncommitted, M1, M2a, M2b, and M2c.

Populations	M0	M1	M2a	M2b	M2c
CD45+CD80++	0.03 ± 0.01	15.76 ± 0.61	0.04 ± 0.01	5.99 ± 0.26	0.06 ± 0.01
CD45+LIGHT+	0.68 ± 0.04	0.99 ± 0.23	0.67 ± 0.06	0.76 ± 0.04	0.62 ± 0.21
CD45+CD206++	0.85 ± 0.08	0.00 ± 0.00	26.13 ± 1.03	0.03 ± 0.01	2.00 ± 1.00
CD45+CD163++	0.77 ± 0.06	0.04 ± 0.01	0.53 ± 0.04	0.25 ± 0.05	12.63 ± 1.52

Definition of abbreviations: M0, uncommitted macrophages; M1, macrophages induced by LPS and IFNr; M2a, macrophages induced by IL-4; M2b, macrophages induced by LPS and IL1b; M2c, macrophages induced by IL-10. Data are representative of the percentage of positive cells among total live CD45+ cells. Blood samples are from at least ten individual donors. Data are shown in mean ± SD.

**Table 2 cells-11-02027-t002:** MFI of CD206.

MFI	FMO	con-HeLa	si-HeLa	con-Siha	si-Siha
CD206	164 ± 23	6741 ± 213	3921 ± 796	3862 ± 116	2885 ± 491

Definition of abbreviations: MFI, mean fluorescence intensity; FMO: Fluorescence minus one; con-HeLa, con-Siha: TAMs macrophages co-cultured with small interfere RNA of negative control treated HeLa cells, and Siha cells, respectively; si-HeLa, si-Siha: TAMs macrophages co-cultured with small interfering RNA of CCL22 treated HeLa cells and Siha cells, respectively; data are representative of mean fluorescence intensity of total live CD45+ cells from at least three independent experiments. Mean ± SD are shown.

**Table 3 cells-11-02027-t003:** MFI of CD206.

MFI	FMO	C-HeLa	T-HeLa	C-Siha	T-Siha
CD206	164 ± 23	1736 ± 97	5846 ± 196	1658 ± 159	6208 ± 617

Definition of abbreviations: MFI, mean fluorescence intensity; FMO: Fluorescence minus one; C-HeLa, C-Siha: TAMs macrophages co-cultured with blank vectors treated HeLa cells, and Siha cells, respectively; T-HeLa, T-Siha: TAMs macrophages co-cultured with vectors of CCL22 treated HeLa cells and Siha cells, respectively; data are representative of mean fluorescence intensity of total live CD45+ cells from at least three independent experiments. Mean ± SD are shown.

## Data Availability

Not applicable.

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
