# Peer review of "CCL22-Polarized TAMs to M2a Macrophages in Cervical Cancer In Vitro Model"

_cells, 2022, doi:10.3390/cells11132027_

Round 1

Reviewer 1 Report

I read with a great interest the manuscript titled “CCL22 polarize TAMs to M2a macrophages in a cervical cancer 2 in vitro model.” This work describes the importance of M2 macrophages in cervical cancer development to identify these cells as a possible therapeutic target of cervical cancer. In general, it is very interesting work and deserves to be published, as TAMs were described as able to modulate the tumour response to cancer-specific treatment in many cancers. As a consequence of their activity in promoting tumour immunosuppressive microenvironment and facilitating metastasis, TAMs may become promising therapeutic targets for cancer treatment. The modulation of their response may facilitate successful immunotherapy as it was investigated in a sorafenib-resistant tumoral cell model (Mills et al., 2016; Mills & Ley, 2014).

In my opinion, this manuscript contains an important message that has been well formulated and it may be published without any changes.

However, I would like to send to the authors the following suggestions for future research on TAMs M2a in cancer:

1) The association of a local infiltration by M2-TAMs with a poor response to chemotherapy in different cancers:

  • 210 patients with oesophageal cancer (Pei, Sun, Zhang, Wang, & Ren, 2014)
  • 417 patients with NSCLC (Sugimura et al., 2015). In NSCLC with EGFR mutation, the number of M2-polarized TAMs also correlated with the decreased responsiveness to EGFR-TKI treatment, as described by Chung et al. in a cohort of 107 patients (Chung et al., 2012).
  • TAMs in pancreatic adenocarcinoma stimulated cancer cells to express high levels of cytidine deaminase, which reduces the sensitivity of cancer cells to chemotherapy by catabolization of the bioactive form of gemcitabine (Amit & Gil, 2013);
  • in breast cancer, TAMs was associated with chemotherapeutic drug resistance and their number in tumoural tissue correlate with the resistance to tamoxifen (Xuan et al., 2014; C. Yang et al., 2015).

2) Various studies found a correlation between high M2-TAM numbers and poor tumour responses to irradiation, provoking regrowth of mammary tumours after radiotherapy both in murine models (Milas, Wike, Hunter, Volpe, & Basic, 1987) and clinical studies (Shiao et al., 2015; Teresa Pinto et al., 2016).

Increasing infiltration by TAMs were associated with an unfavourable outcomes after radiotherapy as they are capable of restore tumour vascularisation and provoke radio-resistance recruiting more monocytes to the site of irradiation (Russell & Brown, 2013). The tissue observations after irradiation showed that DNA damage, cell death and increased tumour hypoxia promoted production of VEGF, SDF-1, and CSF-1, resulting in the recruitment, infiltration, and retention of monocytes/macrophages within the tumour (Russell & Brown, 2013). The recruited TAMs were heterogeneous and able to release different cytokines and metalloproteinase to promote blood vessel formation within the tumour (Q. Guo et al., 2016).

Author Response

Cover letter to reviewer 1

Thank you for your inspiring comments concerning our manuscript entitled “CCL22 polarize TAMs to M2a macrophages in a cervical cancer in vitro model”. (ID: cells-1731814). We have studied comments carefully and think further.

Suggestion 1): The association of a local infiltration by M2-TAMs with a poor response to chemotherapy in different cancers:

210 patients with oesophageal cancer (Pei, Sun, Zhang, Wang, & Ren, 2014)

417 patients with NSCLC (Sugimura et al., 2015). In NSCLC with EGFR mutation, the number of M2-polarized TAMs also correlated with the decreased responsiveness to EGFR-TKI treatment, as described by Chung et al. in a cohort of 107 patients (Chung et al., 2012).

TAMs in pancreatic adenocarcinoma stimulated cancer cells to express high levels of cytidine deaminase, which reduces the sensitivity of cancer cells to chemotherapy by catabolization of the bioactive form of gemcitabine (Amit & Gil, 2013);

in breast cancer, TAMs was associated with chemotherapeutic drug resistance and their number in tumoural tissue correlate with the resistance to tamoxifen (Xuan et al., 2014; C. Yang et al., 2015).

Reply to suggestion 1: Thank you very much for the feasible suggestion! Your suggestion remind us to avail clinical data sufficiently in the future. Though we have focused on the association between overall survival rate and the local infiltration of TAMs, it is valuable to study the association between outcome of chemotherapy and TAMs infiltration.

Suggestion 2): Various studies found a correlation between high M2-TAM numbers and poor tumour responses to irradiation, provoking regrowth of mammary tumours after radiotherapy both in murine models (Milas, Wike, Hunter, Volpe, & Basic, 1987) and clinical studies (Shiao et al., 2015; Teresa Pinto et al., 2016).

Increasing infiltration by TAMs were associated with an unfavourable outcomes after radiotherapy as they are capable of restore tumour vascularisation and provoke radio-resistance recruiting more monocytes to the site of irradiation (Russell & Brown, 2013). The tissue observations after irradiation showed that DNA damage, cell death and increased tumour hypoxia promoted production of VEGF, SDF-1, and CSF-1, resulting in the recruitment, infiltration, and retention of monocytes/macrophages within the tumour (Russell & Brown, 2013). The recruited TAMs were heterogeneous and able to release different cytokines and metalloproteinase to promote blood vessel formation within the tumour (Q. Guo et al., 2016).

Reply to suggestion 2: Thank you for the detailed suggestion! As a student major in Obstetrics and Gynecology, I am not familiar with the study in radiotherapy. However, it is a great idea to cooperate with people working in radiotherapy. I will read and learn from those papers carefully.

Thank you again for your comments and suggestions!

Best regards,

Qun Wang

Department of Obstetrics and Gynecology,

University Hospital, LMU Munich,

81377 Munich, Germany

[email protected]

Reviewer 2 Report

In the study, Wang and colleagues revealed that CCL22 was secreted by tumor associated macrophages (TAMs) in vitro model. They also found that CD80++ was the marker of M1 and M2b, CD206++ as marker of M2a, and CD163++ as marker of M2c, compared with M0 macrophages. Although the manuscript is potentially interesting, the problem with this study is that it is rather premature, inferential, and observational. The statement that CCL22 polarizes TAMs to M2a macrophages is not firmly established by the present data. Specific comments are as follows.

Major points.

  1. The authors claim that CCL22 polarizes TAMs towards M2a macrophages in cervical cancer via an autocrine pathway. In Figure 3, however, the effect of siCCL22 on CD206 expression is subtle and it is unclear that such a small change could have a physiological role. The effect of CCL22 on CD206 expression should be validated by other assays. To further support the authors' findings, it is necessary to determine whether overexpression of CCL22 by expression vectors increases CD206 levels.

  1. In Figure 1, it is assumed that M0 macrophages cocultured with HeLa or Siha cells were polarized to M2a phenotype. Because the authors claim that CD206++ was the marker of M2a, the expression levels of CD206 should be compared each other as well to support the authors’ conclusion.

Minor points.

  1. Figure 2a, c, e, and g: The overall quality of the pictures (dot plots) is too low, and the sizes are too small.
  2. Figure 1: The expression level of CCL22 in M0 macrophages should be presented.
  3. English should be carefully revised by a native English speaker or a professional English editing service.

Author Response

Cover letter to reviewer 2

Thank you for your precious comments concerning our manuscript entitled “CCL22 polarize TAMs to M2a macrophages in a cervical cancer in vitro model” (ID: cells-1731814). We have studied comments carefully and made the following revisions. It will be appreciated to take your time to read and appraise them.

Major point 1: The authors claim that CCL22 polarizes TAMs towards M2a macrophages in cervical cancer via an autocrine pathway. In Figure 3, however, the effect of si-CCL22 on CD206 expression is subtle and it is unclear that such a small change could have a physiological role. The effect of CCL22 on CD206 expression should be validated by other assays. To further support the authors' findings, it is necessary to determine whether overexpression of CCL22 by expression vectors increases CD206 levels.

Reply to major point 1: Thank you so much for the valuable comments. Indeed, the effect of si-CCL22 on CD206 looks subtle, yet it is lucky for us that it is a result of significantly difference. We performed the independent experiments at least 10 times. The trends were stable. Your suggestion of using vectors can totally elevate the quality of this paper to make it more logically complete. We have performed experiments to upregulate the expression of CCL22. We found that after transfection CCL22-vectors, CD206 in TAMs co-cultured with HeLa and SiHa cells were significantly elevated. Accordingly parts in abstract, results, and methods have been added as followed:

Abstract: Line30-32: By regulating CCL22, we found that mean fluorescence intensity (MFI) of CD206 in TAMs was significantly affected compared with the control group.

Results: Line 164-169: The efficiency of CCL22 vectors was confirmed by real-time PCR analyses (Fig. 3f). The expression of CCL22 mRNA in T-CCL22 TAMs were isolated form the co-cultured cells. We next detected the expression of CD206 of T-CCL22-TAMs compared to con-TAMs by flowcytometry. As displayed in Fig. 3ghij and Table 3, over-expressed CCL22 greatly ele-vated the MFI of CD206 of TAMs co-cultured with HeLa and Siha cells, respectively.

Method: We borrowed the plasmid expressing CCL22 from the group of Prof. David Anz (Univer-sity Hospital of Munich, LMU, Munich, Germany) containing a human CCL22 isoform (NM-002990) coding sequence (CDS: CATGGATCGTACAGACTCATCCTGGTGTCCTCGTCCTCCTGCTGTGGCGCTTCACAACTGAGGCAGGCCCCTACGGCGCCAACATGGAAGACAGCGTCTGCTGCCGTGTATCCGTTACCGTCTGCCCCTGCGCGTGGTGAAACACTTCTACTGGACCTCAGACTCCTGCCCGAGGCCTGGCGTGGTGTTGCTAACCTTCAGGTAAGGGATCTGTGCCGATCCCAGAGTGCCCTG) directly cloned into pmx-vector. They confirmed all constructs via sequencing.

Major point 2: In Figure 1, it is assumed that M0 macrophages cocultured with HeLa or Siha cells were polarized to M2a phenotype. Because the authors claim that CD206++ was the marker of M2a, the expression levels of CD206 should be compared each other as well to support the authors’ conclusion.

Reply to major point 2: Thank you for your careful comment. We have revised the line 114 to 117 from ” M2a macrophages were found to have significantly higher percentage of CD206++ compared to M0 macrophages (Fig. 2cd). LIGHT showed no significantly change (Fig. 2ef). M2c macrophages were found to have significantly higher amounts of CD163++ compared to M0 macrophages (Fig. 2gh).” to “M2a macrophages were found to have significantly higher percentage of CD206++ compared to all other subtypes of macrophages (Fig. 2cd). LIGHT showed no significantly change (Fig. 2ef). M2c macrophages were found to have significantly higher amounts of CD163++ compared to all other subtypes of macrophages”.

Minor point 1: Figure 2a, c, e, and g: The overall quality of the pictures (dot plots) is too low, and the sizes are too small.

Reply to minor point 1: Thank you for the advice! We have improved the quality of pictures to 600 dpi, and magnified the pictures.

Minor point 2: Figure 1: The expression level of CCL22 in M0 macrophages should be presented.

Reply to minor point 2: Thank you for the suggestion! We agree with you, too. However, as the figure showed below, the expression of CCL22 in M0 macrophages was significantly lower than M2a macrophages, and significantly higher than M1, M2b, and M2c macrophages. Moreover, the expression of CCL22 in TAMs co-cultured with HeLa or SiHa cells was significantly higher than M1 macrophages, and significantly lower than M0 macrophages. The reason of CCL22 in TAMs to be lower than M0 macrophages is that TAMs is a collection of all subtypes of macrophages but not only M2a macrophages. However, it is a little complicated to explain. Since the expression of CCL22 in TAMs is significantly higher than M1, it can suggest that CCL22 may acts in cervical tumor progression. In addition, CCL22 has been proved to be higher in cervical cancer tissue compared to normal part (PMID: 28086903). Could we use the original figure?

Minor point 3: English should be carefully revised by a native English speaker or a professional English editing service.

Reply to minor point 3: Thank you for the advice. We have searched help from a professional English editing service. The revised parts are high lightened by yellow. We hope it will hit the demand of the journal.

Thank you again for your comments and suggestions!

Best regards,

Qun Wang

Department of Obstetrics and Gynecology,

University Hospital, LMU Munich,

81377 Munich, Germany

[email protected]

Round 2

Reviewer 2 Report

The authors have addressed almost all of my concerns. However, figure legends should be corrected (e.g. Figure 3).

Author Response

Cover letter to reviewer

Thank you for your precious comments concerning our manuscript entitled “CCL22 polarize TAMs to M2a macrophages in a cervical cancer in vitro model” (ID: cells-1731814). We have studied comments carefully and made the following revisions. It will be appreciated to take your time to read and appraise them.

Minor point 1: The authors have addressed almost all of my concerns. However, figure legends should be corrected (e.g. Figure 3).

Reply to minor point 1: Thank you for the advice! We are sorry for the careless! We have revised the uppercase letters into lower case letters legend to make letters in figures to be consistent with those in figure legends. We have made up the missing information in figure 3 as followed:

Figure 3. CCL22 could polarize TAMs towards M2a macrophages in cervical cancer via an autocrine pathway. (a) Total RNA from si-CCL22-TAMs and con-TAMs was analyzed by real-time PCR in triplicate for CCL22 mRNA expression. (b, d) Flowcytometric analysis of MFI (mean fluorescence intensity) of CD206 on the surface of si-CCL22 TAMs and con-TAMs co-cultured with HeLa and Siha cells. (f)Total RNA from CCL22-vectors treated TAMs and con-TAMs was analyzed by real-time PCR in triplicate for CCL22 mRNA expression. (g, i) Flowcytometric analysis of MFI (mean fluorescence intensity) of CD206 on the surface of CCL22-vectors treated TAMs and con-TAMs co-cultured with HeLa and Siha cells. Red line represents MFI of FMO CD206 group; blue line represents MFI of control CCL22 group; orange line represents si-CCL22 or CCL22-vectors treated group. Bar graphs (c, e, h, j) represent the mean and SD of MFI of CD206 in treated macrophages co-cultured with cervical cancer cell line HeLa and Siha respectively, for at least three independent experiments.  * Represents p<0.05

Thank you again for your comments and suggestions!

Best regards,

Qun Wang

Department of Obstetrics and Gynecology,

University Hospital, LMU Munich,

81377 Munich, Germany

[email protected]